# Association between Social Support and Physical Activity in Patients with Coronary Artery Disease: Multiple Mediating Roles of Self-Efficacy and Autonomous Motivation

**DOI:** 10.3390/healthcare10030425

**Published:** 2022-02-24

**Authors:** Nam-Sin Han, Mi-Hwa Won

**Affiliations:** 1Department of Nursing, Wonkwang University Hospital, Iksan 54538, Korea; namsinh@wkuh.org; 2Department of Nursing, Wonkwang University, Iksan 54538, Korea

**Keywords:** confidence, exercise, motivation, perceived environment

## Abstract

Physical inactivity in patients with coronary artery disease is linked to recurrent cardiac events. Given that social support may be an enduring major factor for physical activity, the mechanism underlying the multiple mediating effects of self-efficacy and autonomous motivation on the relationship between social support and physical activity in patients with coronary artery disease has hardly been examined. Therefore, this study aimed to clarify the multiple mediating roles of social support and physical activity on the relationship between self-efficacy and autonomous motivation in patients with coronary artery disease. This descriptive cross-sectional study included 190 inpatients who were diagnosed with coronary artery disease and admitted to a cardiology ward university hospital in Korea. Parallel multiple mediated models were tested using the SPSS PROCESS macro. The direct effects of social support on physical activity and the indirect effects of self-efficacy and autonomous motivation on social support and physical activity were statistically significant. Thus, positive social support from health-care providers has an important role to play in promoting physical activity by increasing self-efficacy and autonomous motivation for physical activity in patients with coronary artery disease.

## 1. Introduction

Coronary artery disease (CAD) is a chronic condition in which the buildup of plaque in coronary arteries causes narrowing or blockages [1]. Globally, CAD is one of the major causes of mortality, which is steadily increasing due to an increase in the elderly population and an unhealthy lifestyle [1,2]. From 2015 to 2019, the annual prevalence of CAD increased by approximately 4.1%, and the total medical costs increased by an average of 10.5% per year in Korea [3]. Despite significant developments in medical therapy and technology, the annual mortality rate of CAD is reported to be approximately five to six times higher than that of those without CAD [2]. In addition, in Korea, it was reported that the CAD mortality rate increased by 0.7%, from 26.7% in 2019 to 27.4% in 2020 [3]. Remarkably, approximately 40% of patients with CAD experience serious cardiac events, such as cardiac ischemia, acute myocardial infarction, and cardiac death [4]. Thus, as the burden of CAD increases with a high risk of recurrence, it is important to identify factors affecting secondary prevention in patients with CAD [3]. Furthermore, it is necessary to investigate ways to maintain secondary prevention behaviors.

Regular physical activity can reduce the prevalence and risk of mortality of CAD. In addition, it is a major determinant of clinical outcomes in patients with CAD [5]. Some studies have found that physical activity is an effective method for improving quality of life by improving cardiopulmonary function through a reduction of high blood pressure, weight, and total cholesterol [6,7,8]. A meta-analysis of cardiac rehabilitation reported that physical activity-enhancing interventions are effective at initiating and maintaining physical activity in patients with CAD [9]. Moreover, some previous studies have demonstrated that regular physical activity and exercise also increase maximum oxygen consumption in the general population and in those with cardiac pathologies and therefore, functional capacity. In this regard, scientific evidence has consistently demonstrated that maximum oxygen consumption enhancement is directly related to an increase in life expectancy of 12–20% for each 3.5 mL/kg/min extra attained [10,11]. Although the results of previous studies revealed various benefits of physical activity for patients with CAD, it was found that the level of physical activity in patients with CAD did not meet the level recommended by the American College of Cardiology and American Heart Association guideline [5,12]. Thus, it is not only important to meet the international recommendations for maintaining or improving fitness in patients with CAD, but also to reduce sedentary behavior and increase physical activity levels of any intensity (i.e., light, moderate, and/or vigorous) [5,12]. For this reason, to continuously adhere to physical activity, which is a factor influencing the disease process and prognosis of patients with CAD, social support, such as increased interest and encouragement from health-care providers, is required [13].

Social support is an essential predictor of physical activity in patients with CAD [14]. Recently, social support has been suggested as a means to decrease adverse health-related outcomes, including morbidity and mortality, because it promotes physical activity in patients with CAD [15]. Moreover, some researchers have shown that individuals who perceive high social support from health-care providers and families participate in higher levels of physical activity [16,17]. Moreover, a randomized controlled study reported improvements in physical capacity and exercise performance in patients with CAD who received a support-based intervention program for four months after the end of a cardiac rehabilitation program [11]. Thus, social support is a contributing factor to correcting the perception of lifestyle modifications, including physical activity, and implementing health behaviors in patients with CAD [18]. Consequently, for patients with CAD to adhere to a physical activity program, tailoring of education and support from health-care providers is required in consideration of their interest in physical activity and their process regarding their recovery [14,18]. Self-efficacy is defined as individuals’ ability to organize their beliefs, which is demonstrated to be one of the predictors of physical activity in patients with chronic diseases [19,20]. Particularly, self-efficacy for physical exercise refers to the ability to exercise in relation to the external and internal barriers of physical activity, such as feeling tired, running out of time, or a lack of community facilities and parks [20]. A previous study reported that self-efficacy was positively related with physical activity in patients with CAD [13]. Longitudinal research has shown that self-efficacy regarding physical activity was a long-term predictor of cardiovascular events during a 13-year follow-up in middle-aged men [21].

Motivation is a decisive factor in behavior, and the autonomous motivation of an individual is the core of a health behavior change strategy [22,23]. Namely, autonomous motivation reflects an individual’s interest and enjoyment with regards to a pursued goal, compared to the controlled motivation involved in behaviors that reward or avoid feelings of guilt [24]. As an optimal and effective strategy for secondary prevention to reduce the risk of CAD recurrence, motivation to implement changed behavior is required [25]. Knittle et al. [23] found that the autonomous motivation of patients with chronic diseases has a positive effect on physical activity and health behavior. In a longitudinal study of patients undergoing cardiac rehabilitation, the higher the individual’s autonomous motivation for physical activity, the higher the level of physical activity adherence, and autonomous motivation was found to be directly related and an indirect relationship between self-efficacy and physical activity was observed [26].

Although previous studies on physical activity in patients with CAD have been conducted, a low level of physical activity and ongoing problems regarding adherence to physical activity are still being reported [27,28]. Thus, insufficient studies have explored potential mechanisms underlying the association between social support, self-efficacy, autonomous motivation, and physical activity in patients with CAD, and further investigation is required. Furthermore, limited data are available concerning the mechanism between these factors in patients with CAD in Korea. Therefore, this study aimed to clarify the multiple mediating roles of self-efficacy and autonomous motivation in the relationship between social support and physical activity in patients with CAD.

## 2. Materials and Methods

### 2.1. Study Design and Sample

Patients diagnosed with CAD at a university hospital with more than 700 beds in Jeollabuk-do, Korea, and admitted to the cardiology ward were included as an accessible population. Convenience sampling was used to recruit participants from April to September 2021. The inclusion criteria were those diagnosed with CAD and admitted to the cardiology ward, aged above 20 years, able to communicate and understand Korean, and agreement to participate in the study. The exclusion criteria were those diagnosed with stroke, Alzheimer’s disease or vascular dementia, memory impairment, psychiatric disorders, end-stage kidney disease, or advanced cancer.

The sample size was calculated using G*Power 3.1.9.2. Considering the median effect size = 0.15, significance level (α) = 0.05, power (1 − β) = 0.90, and 18 independent variables, the required number of samples was calculated to be 183. Considering the dropout rate, 200 patients with CAD were recruited, and a total of 190 patients were included for data analysis, excluding 10 questionnaires with poor or missing data.

### 2.2. Ethical Considerations

This study was conducted after obtaining approval from the institutional review board (IRB) of the hospital to which the primary researcher belongs. Written informed consent was obtained from patients with CAD after the purpose and contents of the study were explained. In addition, based on the general data protection regulation guideline, anonymity and confidentiality of the information gathered from participants were guaranteed and participation was completely voluntary. To protect the personal and sensitive information of the participants in the structured questionnaire, sentences about personal information were minimized.

### 2.3. Instruments

#### 2.3.1. Sociodemographic and Clinical Characteristics

The characteristics of the patients were constructed based on a literature review [13,22,26], including age, gender, education level, living arrangement, employment, body mass index (BMI), periods since diagnosis, left ventricular ejection fraction (LVEF), percutaneous cardiac intervention (PCI), number of chronic diseases, and prescribed cardiac medications.

#### 2.3.2. Social Support

The Health Care Climate Questionnaire (HCCQ) was used to assess the degree of perception of an individual’s perceived social support of their health-care providers in health-care environments [29]. The HCCQ consists of a total of 15 items rated on a 7-point Likert scale, ranging from 1 point for ‘not at all’, 4 points for ‘average’, and 7 points for ‘strongly agree’, and 1 negative question is inversely converted. The score distribution ranges from a minimum of 15 to a maximum of 105 points, and a higher score indicates higher perceived social support. The Cronbach’s alpha for the original scale was 0.89 [29], and the reliability of this study was 0.91.

#### 2.3.3. Self-Efficacy

The Cardiac Exercise Self-Efficacy Scale (CESS) was used to measure individuals’’ confidence in their ability to perform physical exercise in a population with cardiac conditions [30]. The CESS consists of a total of 16 items rated on a 5-point Likert scale, ranging from 1 point for ‘not at all true’ to 5 points for ‘completely true’. The score distribution ranged from 16 to 80 points, with a higher score indicating higher exercise self-efficacy. The reliability of the original scale was Cronbach’s alpha was 0.90 [30], and the reliability of this study was 0.94.

#### 2.3.4. Autonomous Motivation

The Korean version of the Treatment Self-Regulation Questionnaire (TSRQ) was used to assess the degree to which individuals have autonomous motivation in patients with cardiovascular disease [31]. The original TSRQ scale was developed to measure individual differences in the types of motivation or self-regulation [32]. The TSRQ consists of a total of 15 items with 3 sub-domains: autonomous motivation (8 items), controllable motivation (4 items), and non-motivation (3 items). Eight autonomous motivation items were measured in this study. Each item uses a 7-point Likert scale, ranging from 1 point for ‘not at all’, 4 points for ‘average’, and 7 points for ‘strongly agree’. The score distribution ranges from 8 to 56 points, with a higher score indicating higher autonomous motivation. The Cronbach’s alpha of the original TSRQ scale was 0.80 [32], that of the Korean version of the TSRQ scale was 0.76 [31], and that of this study was 0.92.

#### 2.3.5. Physical Activity

The Korean version of the Global Physical Activity Questionnaire (GPAQ) was used to assess the degree of individual physical activity. The original GPAQ was developed by the WHO to assess information on physical activity participation intensity and duration of three domains (work, transport, and recreation) over the past seven days [33]. Based on the GPAQ guidelines, the total physical activity was calculated using the metabolic equivalent of task-minute per week (MET-min/week): The intensity of each domain (vigorous MET values as 8.0, moderate MET values as 4.0, and transport Met values as 4.0) × the physical activity time (min) × the number of activities per week was calculated. A higher MET-min/week score indicates a higher degree of physical activity.

### 2.4. Data Collection

Eligible participants were recruited through a poster for recruitment of research participants on the bulletin board of the cardiovascular medicine ward. The researcher identified participants using hospital registries and medical records. Two trained researchers obtained written consent from the participants and collected the data using a self-report questionnaire. Interviews took approximately 10–20 min, and clinical data were collected from electronic medical records.

### 2.5. Statistical Analysis

Data analysis was performed using IBM SPSS/WIN statistics program (version 26). The analysis method for each variable was as follows: (1) The patients’ sociodemographic and clinical characteristics, social support, self-efficacy, autonomous motivation, and physical activity were described as a frequency and percentage, mean, and standard deviation. (2) For univariate analysis, an independent sample *t*-test and one-way ANOVA were used, and the correlation between variables was confirmed using Pearson’s correlation. (3) To confirm the multiple mediating effects of self-efficacy and autonomous motivation on the relationship between social support and physical activity in patients with CAD after controlling for covariables, including age, gender, educational level, employment, LVEF, and number of chronic diseases in all models. (4) A parallel multi-intervention model was analyzed using PRESS macro version 3.3.5 (No. 4) of Hayes [34]. In addition, the direct and indirect effects were analyzed using 95% bootstrapped confidence intervals based on 10,000 resamples, and if zero was not included in the 95% confidence interval, the direct and indirect effects were significant.

## 3. Results

### 3.1. Patients’ Sociodemographic and Clinical Characteristics

Among the 190 patients with CAD, the mean age of the patients was 66.95 years (SD = 10.74), and 53.7% were aged 65 years and older. Male (54.7%) and female (45.3%) patients and 40.5% of the patients had less than middle school education. Most patients (78.9%) were living with their families, and 94 (49.5%) were currently employed. The highest proportion of BMI was in obesity (30.5%), followed by overweight (27.4%), normal weight (25.3%), and underweight (16.8%). Half (52.6%) of the patients had less than 1 year since diagnosis. The mean LVEF was 49.05% (SD = 10.43), and 87 patients (42.8%) had more than 50% LVEF. The majority (71.1%) of the patients underwent PCI, and 87 patients (45.8%) responded that they had 1 chronic disease (Table 1). Among the prescribed cardiac medications, the highest percentage was prescribed with lipid-lowering agents (88.7%), followed by clopidogrel (84.0%), aspirin (67.5%), ACEI or ARB (50.5%), and beta-blockers (40.7%).

Physical activity in patients with CAD was statistically different between sexes (t = 5.36, *p* < 0.001) and education levels (F = 14.05, *p* < 0.001), employment (t = 2.07, *p* = 0.039), BMI (t = 3.69, *p* = 0.013), LVEF (F = 18.12, *p* < 0.001), PCI (t = −6.80, *p* < 0.001), and number of diseases (F = 21.05, *p* < 0.001) (Table 1).

### 3.2. Relationship between Social Support, Self-Efficacy, Autonomous Motivation, and Physical Activity in Patients with CAD

The mean (SD) social support, self-efficacy, autonomous motivation, and physical activity of patients with CAD were 81.94 (SD = 14.34), 45.90 (SD = 14.28), 42.30 (SD = 10.45), and 1268.63 (SD = 1054.43) MET-min/week, respectively. Physical activity was found to have a statistically significant positive correlation with social support (r = 0.53, *p* < 0.001), self-efficacy (r = 0.48, *p* < 0.001), and autonomous motivation (r = 0.54, *p* < 0.001) in patients with CAD (Table 2).

### 3.3. Multiple Mediating Effects of Self-Efficacy and Autonomous Motivation on the Relationship between Social Support and Physical Activity in Patients with CAD

Table 3 presents the results of the parallel multiple mediation model used to assess the indirect effects of social support on physical activity through self-efficacy and autonomous motivation. Patients who reported better social support revealed higher levels of self-efficacy (Ba_1_ = −0.33, *p* < 0.001) and autonomous motivation (Ba_2_ = 0.37, *p* < 0.001). Patients who reported higher levels of self-efficacy had higher levels of physical activity (Bb_1_ = 15.02, *p* = 0.005), and patients who reported higher levels of autonomous motivation had higher physical activity (Bb_2_ = 18.50, *p* = 0.008). Additionally, patients who reported better social support reported higher levels of physical activity (Bc = 16.34, *p* < 0.001).

### 3.4. Direct and Indirect Effects on Physical Activity in Patients with CAD

Table 4 shows the direct and indirect effects of social support on physical activity using 10,000 bootstrap samples and a 95% bootstrap confidence interval. The direct effect of social support on physical activity was statistically significant (Bc = 16.34), and the 95% bootstrap confidence interval (6.88 to 26.39) was above zero. The indirect effect of social support on physical activity through self-efficacy was statistically significant (Ba_1_b_1_ = 4.92), and the 95% bootstrap confidence interval (1.89 to 8.78) was above zero. In addition, the indirect effect of social support on physical activity through autonomous motivation was statistically significant (Ba_2_b_2_ = 7.00), and the 95% bootstrap confidence interval (1.69 to 12.37) was above zero. In addition, the total indirect effects were statistically significant (Bc′ = 11.92), and the 95% bootstrap confidence interval (5.82 to 18.30) did not include zero. (Table 4). Figure 1 shows the relationship between social support, self-efficacy, autonomous motivation, and physical activity, which was tested in the same manner as above.

## 4. Discussion

Physical activity in patients with CAD is an important evaluation index for secondary preventive treatment [5]. To the best of our knowledge, this study is the first to investigate the multiple mediating roles of self-efficacy and autonomous motivation on the relationship between social support and physical activity in patients with CAD. Interestingly, this study showed that self-efficacy and autonomous motivation had multiple mediating roles in the relationship between social support and physical activity in patients with CAD. More specifically, social support, self-efficacy, and autonomous motivation were positively associated with physical activity, indicating that higher levels of these variables were related to higher levels of physical activity in patients with CAD.

In this study, the mean physical activity of inpatients with CAD was found to be 1268.63 (SD = 1054.43) MET-min/week. However, our result was low compared to a previous study of community-dwelling patients with CAD in Korea, which reported a mean physical activity of 2042.77 (SD = 4330.38) MET-min/week [35]. Although the results of these studies analyzed patients with CAD over a mean age of 60 years, this inconsistency may be related to differences in the type of measurements of physical activity. Moreover, this is interpreted as a difference depending on the severity of the disease and whether the patients are admitted to the hospital. In this regard, more studies are required to measure physical activity within the CAD population using self-reported scale and objective measurement tools, such as wearable physical activity monitoring devices [25].

Additionally, in this study, there were statistically significant differences regarding physical activity in patients with CAD in terms of gender, education level, employment, BMI, LVEF, PCI, and the number of chronic diseases. These results were similar to those of a study that investigated the differences between patients’ characteristics and physical activity for CAD [35]. Therefore, health-care providers should consider these points when developing programs to promote physical activity in patients with CAD. Future studies should identify patients’ general factors, considering the cultural diversity of health-care and clinical settings that affect the physical activity of patients with CAD.

In parallel multiple mediated analyses, first, the social support of patients with CAD had a significant positive correlation with physical activity. This finding was similar to those of previous studies on CAD, which revealed that patients with higher social support from health-care providers showed higher levels of physical activity [14,17]. This finding is supported by previous studies demonstrating that social support is a strong factor affecting physical activity within the CAD population [14,26]. These results indicate that social support, such as information provision and encouragement by health-care providers, is the driving force behind maintaining and promoting physical activity. Thus, health-care providers should recognize the importance of social support, such as individualized education and encouragement, considering the preferences and goals of physical activity and appropriate feedback to promote physical activity in patients with CAD.

Second, self-efficacy was found to influence physical activity in patients with CAD in this study. This result is in line with the results of a previous study, which reported that perceived self-efficacy was positively associated with long-term adherence to physical activity during cardiac rehabilitation after discharge [22]. Moreover, a previous study showed that self-efficacy could influence adherence to recommended physical activities in patients with CAD [36]. Namely, these results indicate that patients with high self-efficacy may have high confidence in their physical activity for secondary prevention, thereby continuing to change behaviors, such as physical activity. Self-efficacy may increase opportunities for physical activity by overcoming various barriers to physical activity [37]. Thus, health-care providers should pay attention to self-efficacy as a major predictor of behavior change because it works as an independent component that triggers an individual’s activity. In this context, further studies are required to establish self-efficacy-enhancing strategies to initiate and maintain compliance with physical activity for secondary prevention in patients with CAD.

Finally, autonomous motivation was found to influence physical activity in patients with CAD in this study. This is consistent with the results of a previous study [26] that analyzed the pathway between autonomous motivation and physical activity in patients with CAD. Moreover, the vital factor that directly affects health behavior was supported by a previous study that reported autonomous motivation in women with cardiovascular disease [31]. In other words, the higher the autonomy motivation, the greater the autonomy motivation for physical activity and the stronger the will to control behavior and make an effort toward one’s own goal so that continuous physical activity can be maintained. Thus, autonomous motivation is a motivation given by one’s own voluntary will [32], is a prerequisite for starting and maintaining physical activity, and affects the continuous physical activity of patients with CAD [38]. Therefore, it is important to develop a strategy to motivate patients with CAD for continuous physical activity to prevent recurrence. Future studies are required to develop a patient-centered intervention strategy that focuses on individual interests and preferences for physical activity in patients with CAD.

More importantly, our main findings confirmed the multiple mediating roles of self-efficacy and autonomous motivation on the relationship between social support and physical activity in patients with CAD. These results indicate that the positive relationship between social support and physical activity was partly explained by the higher levels of self-efficacy and autonomous motivation. Moreover, our results suggest that to promote physical activity in patients with CAD, the first step should be to build self-efficacy and autonomously motivate patients to engage in physical activity through social support from health-care providers. Thus, health-care providers should be aware that self-efficacy and autonomous motivation for physical activity can be promoted by the health information provider’s autonomy support, such as accepting the individual’s point of view and providing feedback on physical activity in patients with CAD.

This study has limitations. As this study has a cross-sectional design using convenience sampling from a single center, there are limitations in explaining the clear causal relationship between social support, self-efficacy, autonomous motivation, and physical activity and generalizing the research results. Therefore, future multicenter longitudinal studies using larger samples are required. Furthermore, as this study used a self-report tool, the results may have been overestimated or underestimated because of recall bias. Further studies are required to identify distinct levels of physical activity using objective scales in patients with CAD.

## 5. Conclusions

This study aimed to identify the levels of physical activity in patients with CAD and the factors affecting physical activity. This study aimed to identify the multiple mediating roles of social support and physical activity in the relationship between self-efficacy and autonomous motivation in patients with CAD. The results showed that social support of patients with CAD can promote physical activity by mediating self-efficacy and autonomous motivation after adjustment for age, gender, education level, employment, LVEF, and number of chronic diseases. Therefore, to promote physical activity as secondary prevention in patients with CAD, social support, such as the interest and encouragement by health-care providers, and a strategy that achieves self-efficacy and motivates autonomy for physical activity are needed. Moreover, health-care providers should be aware that it is important to reduce sedentary behaviors and replace them with physical activity at any intensity (even light intensity) to increase the total energy expenditure in patients with CAD. Hence, it is necessary to develop and apply an intervention program that enhances self-efficacy and autonomous motivation to promote physical activity in patients with CAD and verify its effectiveness.

## Figures and Tables

**Figure 1 healthcare-10-00425-f001:**
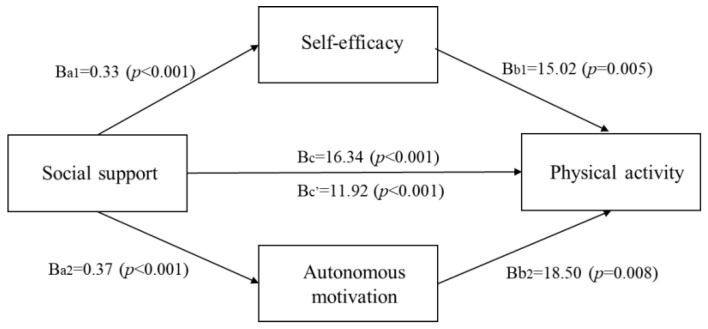
Parallel multiple mediation model testing self-efficacy and autonomous motivation as mediators of social support on physical activity.

**Table 1 healthcare-10-00425-t001:** Differences in physical activity according to sociodemographic and clinical characteristics (N = 190).

Variables	Category	n (%)	PA (MET-In/Week) Mean ± SD	t or F	*p*
Age (years)	<65	88 (46.3)	1357.88 ± 1136.57	1.08	0.28
	≥65	102 (53.7)	1191.62 ± 977.18		
Gender	Men	104 (54.7)	1199.62 ± 632.43	5.36	<0.001
	Women	86 (45.3)	632.43 ± 43		
Education level	≦Middle school	77 (40.5)	526.46 ± 59.99	14.05	<0.001
	High school	75 (39.5)	1208.43 ± 139.53		
	≥College	38 (20.0)	1488.26 ± 1202.96		
Living arrangement	Alone	40 (21.1)	1090.85 ± 867.83	−1.2	0.231
	With family	150 (78.9)	1316.04 ± 1096.55		
Employment	No	96 (50.5)	1109.38 ± 955.22	2.07	0.039
	Yes	94 (49.5)	1126.52 ± 114.97		
BMI (kg/m^2^)	Underweight (<18.5)	32 (16.8)	1044.12 ± 1162.65	3.69	0.013
	Normal (18.5–22.9)	48 (25.3)	1191.86 ± 162.19		
	Overweight (23–24.9)	52 (27.4)	923.19 ± 130.55		
	Obesity (≥25)	58 (30.5)	861.39 ± 117.29		
Periods since diagnosed	<1	100 (52.6)	1183.86 ± 967.65	−1.16	0.244
(years)	≥1	90 (47.4)	1362.82 ± 1141.25		
LVEF (%)	<40	28 (14.7)	863.65 ± 74.33	18.12	<0.001
	41–49	75 (39.5)	933.71 ± 1109.87		
	≥50	87 (42.8)	1759.92 ± 1135.70		
PCI	No	55 (28.9)	1127.50 ± 152.03	−6.8	<0.001
	Yes	135 (71.1)	893.11 ± 703.35		
Number of chronic	1	87 (45.8)	1755.77 ± 1165.30	21.05	<0.001
diseases	2	64 (33.7)	913.25 ± 755.74		
	≥3	39 (20.5)	765.12 ± 702.54		
Prescribed cardiac	Aspirin, yes	131 (67.5)	1288.27 ± 1088.01	0.495	0.622
medications	Clopidogrel, yes	163 (84.0)	1278.99 ± 1035.46	0.5	0.615
	Lipid-lowering agents, yes	172 (88.7)	1261.65 ± 1085.12	−0.03	0.978
	ACEI or ARB, yes	98 (50.5)	1237.83 ± 1019.46	−0.39	0.691
	Beta blocker, yes	79 (40.7)	1352.96 ± 959.90	0.99	0.32

PA = physical activity; MET = metabolic equivalents task; SD = standard deviation; BMI = body mass index; LVEF = left ventricular ejection fraction; PCI = percutaneous cardiac intervention; ACEI = angiotensin-converting enzyme inhibitor; ARB = angiotensin II receptor blocker.

**Table 2 healthcare-10-00425-t002:** Descriptive statistics and correlations among the variables in the study (N = 190).

Variables	Mean ± SD	1	2	3
		r (*p*)	r (*p*)	r (*p*)
1. Social support	81.94 ± 14.34	1		
2. Self-efficacy	45.90 ± 14.28	0.40 (<0.001)	1	
3. Autonomy motivation	42.30 ± 10.45	0.61 (<0.001)	0.39 (<0.001)	1
4. Physical activity (MET-in/week)	1268.63 ± 1054.43	0.53 (<0.001)	0.48 (<0.001)	0.54 (<0.001)

MET = metabolic equivalents task; SD = standard deviation.

**Table 3 healthcare-10-00425-t003:** Mediating effect of self-efficacy and autonomous motivation in the relationship between social support and physical activity (N = 190).

Path	Variables	B	SE	t	*p*	95% CI		Adj. R^2^	F (*p*)
						LL	UL		
a_1_	Social support → Self-efficacy	0.33	0.07	4.67	<0.001	0.19	0.47	0.217	6.27 (<0.001)
a_2_	Social support → Autonomous motivation	0.37	0.04	8.87	<0.001	0.29	0.46	0.459	19.20 (<0.001)
c	Social support → Physical activity	16.34	4.89	3.34	0.001	6.68	25.99	0.541	21.11 (<0.001)
b_1_	Self-efficacy → Physical activity	15.02	4.25	3.53	0.005	6.62	23.41		
b_2_	Autonomous motivation → Physical activity	18.50	6.99	2.64	0.008	4.69	32.31		

SE = standard error; CI = confidence interval; LL = lower level; UL = upper level; Adj. R^2^ = Adjusted R^2^ Adjusted gender, educational level, Employment, BMI, PCI, LVEF, Number of chronic diseases.

**Table 4 healthcare-10-00425-t004:** Direct and indirect effect on physical activity (N = 190).

Variables	Direct Effect				Indirect Effect			
	B	Boot SE	95% CI		B	Boot SE	95% CI	
			Boot LLCI	Boot ULCI			Boot LLCI	Boot ULCI
Social support → Physical activity (c)	16.34	4.97	6.88	26.39				
Social support → Self-efficacy → Physical activity (a_1_b_1_)					4.92	1.78	1.89	8.78
Social support → Autonomous motivation → Physical activity (a_2_b_2_)					7	2.68	1.69	12.37
Total (c′)					11.92	3.15	5.82	18.3

Boot SE = Bootstrap for standard error; CI = confidential interval; Boot LLCI = the lower limit of B in the 95% confidential interval; Boot ULCI = the upper limit of B in the 95% confidential interval.

## Data Availability

Data sharing is not applicable to this article as data analyzed in the current study contains participant medical information. Sharing of this information would breach patient privacy, confidentiality and the ethics approval gained for the study.

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
