# Peer review of "Association between Social Support and Physical Activity in Patients with Coronary Artery Disease: Multiple Mediating Roles of Self-Efficacy and Autonomous Motivation"

_healthcare, 2022, doi:10.3390/healthcare10030425_

Round 1

Reviewer 1 Report

Thank you for the opportunity to review this work. I hope you accept my review, including general comments below in the kind and encouraging manner they are intended.

-Overall, this is a very interesting and much-needed area of research. There are a number of general and often minor details which need to receive attention as set out below:

-Line 13 should read to clarify not to clarifying

-Line 25 – CAD abbreviation provided in abstract

-Authors state in lines 31-32 : …’In addition, in Korea, it was reported that the CAD mortality rate increased by 2.6%, from 26.7% in 2019 to 27.4% in 2020 [3]’…..This is not a 2.6% increase???

-Line 44 -spelling mistake should read researchers

-Under ‘Ethical considerations’ how data handling was protected e.g. GDPR or equivalent procedures should be added.

-Table 1 title- use all or no capital letters (except for ‘and’)-be consistent.

-In table 1 ‘collage’ should read college?

-Table 2- one bold ‘p’ in titles, other ‘p’ are in italics -remove bold font?

-Table 2 title and content do not read accurately- correlation to what? Units of measurement nee dto be provided for each variable for example physical activity has a mean and SD of 1268.63±1054.43…what is this minutes? Seconds? Steps?. Same comment for the whole table

-Check all table titles are consistent with the use of capital letters or not. E.g. Mediating Effect of self-efficacy and autonomous motivation in the relationship between social support and physical activity (N=190)

-Line 295 and 309 need reworded from second and third…does not read well

-Line 350 – is restenosis the most accurate term here?

-Generally, the conclusion is concise which is good, however, it could be slightly more impactful and a minor revision with this in mind is suggested.

Reviewer 2 Report

The authors used adequate methodology and included enough details in different sections with summary tables and figures. Below is minor comment for authors to consider.

Can you please elaborate on how you collect sociodemographic and clinical characteristics? These factors were chosen based on literature review (line 116). However, it’s better to clarify the mode of data collection. Is it through self-reported questioners or interview?

Reviewer 3 Report

The aim of the present study is to investigate the multiple mediating roles of exercise self-efficacy and autonomous motivation on the relationship between social support received from healthcare providers and physical activity levels in patients with coronary artery disease. The main results of this study showed that social support from healthcare providers, exercise self-efficacy and autonomous motivation were positively associated with physical activity levels, and that the role of social support in promoting physical activity was partially explained by higher levels of both exercise self-efficacy and autonomous motivation.

The manuscript is interesting and easy to read. Below, I make some comments and suggestions with the purpose to help authors to improve the quality of their manuscript.

GENERAL COMMENTS:

Please let a native English speaker to review your manuscript.

Keywords: Please avoid to use keywords already included in the title.

Line 55: Replace “…it facilitates physical activity…” for “it promotes physical activity”.

Line 57: Suggest to replace “…have higher levels…” for “perform higher levels”.

Line 63: Please add: “…to adhere to a physical activity program for…”.

Lines 110-112: Please rewrite: “In addition, anonymity and confidentiality of the information gathered from participants were guaranteed and participation was completely voluntary”.

Line 132: Please delete one of the apostrophes at the end of the sentence.

Line 133: Rewrite “… to perform physical exercise in population with cardiac conditions”.

Line 153: Please replace “degree” for “level”.

SPECIFIC COMMENTS:

Introduction section:

I suggest authors to briefly explain the concept of coronary artery disease, since this is a key point in your study.

Lines 39-46: Regular physical activity and exercise also increase maximum oxygen consumption (VO2max) in general population and in those with cardiac pathologies and therefore, functional capacity. In this regard, scientific evidence has consistently demonstrated that VO2max enhancement is directly related to increased life expectancy of 12-20% for each 3.5 ml/kg/min extra attained. I suggest authors to include this idea in this paragraph.

Lines 46-52: In my opinion, it is not only important to meet the international recommendations for maintaining or improving fitness in population with non-transmissible chronic diseases, but also to reduce sedentary behaviour and replace it with, at least, light-intensity physical activity. Scientific evidence has shown important health benefits when weekly energy expenditure is increased via reducing sedentary behaviour and increasing physical activity levels of any intensity (i.e., light, moderate and/or vigorous). In this regard, NEAT (non-exercise activity thermogenesis) plays an important role in weekly energy expenditure. I suggest authors to include this issue.

Lines 61-63: Please rewrite. Difficult to follow.

Lines 67-69: Taking into account that in industrialized countries the majority of people are physically inactive (i.e., not meet the international recommendations of physical activity for maintaining or enhancing healthy fitness), there are different reasons why people do not perform enough/regularly physical activity. In this sense, lack of time is the first reason why people are physically inactive. Furthermore, socio-economic factors like economic incomes, education level, age and gender are factors that affect physical activity and exercise levels. In this sense, it is important to consider the role of active communities, which are communities where is easy to be physically active (e.g., bike/running lanes, sport facilities, parks, etc.), thus promoting physical activity. On the other hand, the lack of this environment might be a barrier to perform physical activity. Suggest to include this idea in the introduction section.

Lines 73-81: Please explain briefly and refer here to motivation’s task-goal (i.e., personal goal and achievement) and ego-goal (i.e., external achievement or social recognition) orientation.

Lines 80-81: How can autonomous motivation directly or indirectly affects physical activity? Please explain.

Materials and Methods:

Lines 157-159: Why did you assess vigorous- and moderate-intensity physical activity and not also light-intensity physical activity? It is well known that light-intensity physical activity (i.e., energetic cost between 1.6 and 2.9 MET) also reports important health benefits and can reduce the incidence of cardiac conditions and associated risk factors. In this regard, it is also important to measure the time dedicated to sedentary behaviours (i.e., those that implies a lack of movement and an energetic cost of ⁓1 to 1.5 MET), since it is considered a risk factor itself.

Results:

Please do not replicate information in text already presented in Tables. I suggest authors to revise this issue and change it accordingly along the results section.

Line 188: Table 1 shows that 49.5% of patients were employed, not 50.5%.

Table 1: Please include the units for the PA variable. What is the reason for dividing your sample above or below 65 years? Why at this age?

Table 3, legend: SE means self-efficacy and also standard error. I suggest to change one of them for a better reader’s comprehension.

Table 4, title: Change “Directing” to “Direct”.

Discussion:

Lines 277-284: You can add here the concept of active communities and discuss the factors that affect individuals to be physically active, like I commented above.

Lines 302-305: Please include in this paragraph the importance of reducing sedentary behaviours and replacing them with physical activity at any intensity (even light-intensity) for increasing total energy expenditure.

Reviewer 4 Report

Congratulations on your work. I find it a very interesting topic and overall very well presented. My criticism is directed to the discussion and conclusions of the study.

I think that there are sentences in which you imply a causal relationship between the variables measured and PA that are not supported by your study results. You would need a longitudinal study to be able to make such statements, so I think you should modify the wording of both parts (discussion and conclusions).

On the other hand, in the tables of results it would be good if you would point out the results of p<0.05. This way, even if it is already indicated in the text, it is easier to quickly identify the significant results. You could also add that PA is in METs per minute per week.
